# Prevalence of VZV Reactivation and Effectiveness of Vaccination with Recombinant Adjuvanted Zoster Vaccine in Allogeneic Hematopoietic Stem Cell Recipients—A Single-Center Analysis

**DOI:** 10.3390/idr17030048

**Published:** 2025-05-02

**Authors:** Ewa Karakulska-Prystupiuk, Magdalena Feliksbrot-Bratosiewicz, Maria Król, Agnieszka Tomaszewska, Wiesław Wiktor Jędrzejczak, Grzegorz Władysław Basak

**Affiliations:** Department of Hematology, Transplantation and Internal Medicine, Medical University of Warsaw, Banacha 1a Str., 02-097 Warsaw, Poland; magdalena.feliksbrot-bratosiewicz@wum.onmicrosoft.com (M.F.-B.); makroll@post.home.pl (M.K.); agnieszka_tomaszewska@onet.eu (A.T.); wieslaw.jedrzejczak@wum.edu.pl (W.W.J.); grzegorz.basak@wum.edu.pl (G.W.B.)

**Keywords:** recombinant adjuvanted herpes zoster vaccine (RZV), herpes zoster (HZ), allogeneic hematopoietic stem cell transplantation (allo-HSCT), secondary immunodeficiency (SID)

## Abstract

Background: Secondary immunodeficiencies in allo-HSCT (allogeneic hematopoietic stem cell transplantation) recipients increase the risk of viral reactivation, making vaccinations a vital issue. There is a paucity of data on the use of recombinant vaccine against herpes zoster (RZV) after allo-HSCT. Methods: This analysis included 149 recipients of allo-HSCT, transplanted in 2012–2022, mainly due to hematological malignancies (>95%). RZV was used from 2021 to 2023 according to the current recommendations of ACIP. The ELISA method was used to assess the VZV IgG antibody titers. Results: VZV reactivation was diagnosed in 49 out of 149 (33%) patients before vaccination, including 5 (3%) patients with reactivation within the first year after transplantation and the remaining 44 (30%) within the subsequent three years. At that time, the majority of patients were not receiving acyclovir prophylaxis. The most common clinical manifestation of reactivation was involvement of intercostal nerves, diagnosed in 40 (81%) patients. Twenty-one recipients (median age: 41) received two doses of RZV (at a median time of 34 months after transplantation, range 12–84 months), the majority of them at an interval of 1 month. The serological post-vaccination response was confirmed in 12 recipients, with a ratio of 2.38–8.3 (median 5.095). The median number of total CD3+CD4+cells in vaccinated patients was 451/μL. Despite vaccination, four patients (19%, three with confirmed serological response) developed herpes zoster. Conclusions: Herpes zoster occurred mainly in the late period after allo-HSCT after completion of acyclovir prophylaxis in over 30% of recipients. The preliminary results indicate that RZV vaccination after allo-HSCT was safe and more than 80% effective at preventing HZ, but some vaccinated individuals did experience HZ.

## 1. Introduction

Recipients of allogeneic hematopoietic stem cell transplantation (allo-HSCT), especially those with graft-versus-host disease (GvHD), constitute a unique group of patients with secondary immunodeficiency (SID). Ablation of the hematopoietic system and damage to lymphopoietic organs due to toxic conditioning chemo-radiotherapy before transplantation lead to the impairment of cellular and humoral immunity. Restoring the total efficiency of the immune system is a multi-stage process spread over time. Cellular immunity deficiencies in allo-HSCT recipients significantly increase the risk of viral infections.

Varicella zoster virus (VZV) is a human alpha herpes virus surviving latent in ganglionic neurons and reactivating to produce herpes zoster (HZ, shingles). According to the literature data, over 90% of the world population harbors latent VZV [1]. At least 50% of infected individuals will reactivate this virus by 85 years of age to develop zoster [2]. During reactivation, VZV spreads transaxonally, but in immunosuppressed patients, may also be detected in peripheral blood mononuclear cells(PBMCs), which promote its dissemination [1,2,3,4,5]. Long-lasting postherpetic neuralgia may be an essential complication in many recipients [1]. The first objective of our study was to assess the incidence of HZ in recipients after HSCT.

Vaccination is a vital way to prevent symptomatic infection in HSCT recipients. However, the response to immunization in these SID patient populations may not be sufficient. There are two zoster vaccines: a live-attenuated vaccine, which is contraindicated in this setting, and the recombinant glycoprotein E vaccine. The efficacy of the recombinant HZ vaccine after autologous stem cell transplantation has been documented [6], but the data on its use after allo-HSCT are still limited. The Advisory Committee on Immunization Practices (ACIP) does not explicitly comment on these patients, choosing to await additional information [6,7,8,9,10,11,12]. Therefore, we considered it important to report our experience with recombinant vaccine against herpes zoster in this patient population.

## 2. Materials and Methods

### 2.1. Study Population

This is a retrospective analysis of patients after allo-HSCT was performed in the years 2012–2022 who remained under the care of the Outpatient Service in the single transplantation center. All patients signed written consent to receive vaccinations. The follow-up time for VZV reactivation in the unvaccinated patients was the first eight years after transplantation.

### 2.2. Allo-HSCT

The type of conditioning was chosen at the responsible physician’s discretion and depended on the underlying hematological disease. Immunosuppressive treatment (GvHD prophylaxis) was a combination of a calcineurin inhibitor (cyclosporin or tacrolimus) and an antiproliferative drug—a short course of either methotrexate or mycophenolate mofetil. All patients with unrelated or mismatched donors received anti-T-cell globulin (2.5–5 mg/kg daily) in a conditioning regimen for 2–3 days prior to transplantation. Diagnosis and grading of acute and chronic GvHD were performed based on clinical symptoms and/or biopsies according to established criteria. Grading of acute GvHD was performed according to the Glucksberg score, while the severity of chronic GvHD was determined according to the National Institutes of Health (NIH) Consensus Criteria 2014 [13,14,15].

All patients received anti-infective prophylaxis, including prophylactic antiviral treatment with acyclovir (Hasco-Lek, Wrocław, Poland) (400–800 mg twice daily) or with letermovir (Merck Sharp & Dohme B.V., Haarlem, The Netherlands) (240–480 mg daily) for the first 100 days after allo-HSCT followed by acyclovir (in doses as above) or valganciclovir (Accord Healthcare, North Harrow, Great Britain) (in the case of CMV reactivation, in doses adjusted for creatinine clearance). Immunosuppressive therapy was discontinued after 6–8 months following allo-HSCT if there was no significant GvHD. All patients who have been receiving immunosuppressive therapy due to cGvHD were advised to continue acyclovir treatment [7,12].

### 2.3. Types and Doses of Vaccine

Recombinant adjuvanted herpes zoster vaccine (RZV) was used in 2021–2023 in patients according to the current recommendations of the Advisory Committee on Immunization Practices (ACIP) for Immunocompromised Adults aged ≥19 years [10]. Vaccination included two doses of the vaccine given 1–2 months apart. The follow-up for efficacy was 1–3 years [8,9,10,11].

### 2.4. Methods

The anti-VZV (IgG) ELISA test(Euroimmun, Lübeck, Germany) was primarily used, with the result expressed as a ratio (the optical density (OD) of the patient sample divided by the OD of the cut-off). A ratio ≤ 0.8 was considered negative, 0.8–1.1 equivocal, and >1.1 positive. A direct chemiluminescent immunoassay (CLIA) by DiaSorin (analyzer LIAISON XL) (DiaSorin, Saluggia, Italy) was used in three patients, with quantitative results expressed in international units per milliliter (IU/mL). A titer ≥ 150 mIU/mL was considered a positive result. The comparison of the results of these two tests may only be qualitative because the ELISA ratio is a relative value and is not standardized to the WHO international unit. Peripheral blood lymphocyte subpopulations were analyzed using flow cytometry. The reference intervals were 309–1139 cells/μL for CD3+4+ cells, 137–823 cells/μL for CD3+8+ cells, 70–460 cells/μL for NK cells, and 80–430 cells/μL for CD19+ B-lymphocytes, and 1.0–5.0 for the CD4+ to CD8+ ratio. Post-vaccination complications were graded according to CTCAE5.0 criteria. Patients gave written consent to the intervention.

## 3. Results

### 3.1. Patients

The analysis included 149 patients, including 85(57%) males, with a median age of 47 years (range 18–73) at allo-HSCT. For twelve patients, this was their second allo-HSCT. The most prevalent diagnoses were acute myeloid leukemia (AML)—55%—then myelodysplastic syndrome (MDS) and acute lymphoblastic leukemia (ALL)—both 11.5%.

The patients’ baseline characteristics are shown in Table 1.

### 3.2. Transplantations

HLA-identical siblings were used for 49 (33%) patients, matched unrelated donors for 83 (56%), mismatched unrelated donors for 12 (8%), and haploidentical related donors for 5 (3%). In total, 104 (70%) patients received myeloablative conditioning (MAC), 40 (27%) reduced-intensity conditioning (RIC), and 5 (3%) non-myeloablative conditioning (NMA).

### 3.3. GvHD

A total of 64 (43%) patients suffered from acute and 85 (57%) from chronic GvHD. Altogether, 67 (45%) patients required chronic immunosuppressive therapy (mostly calcineurin inhibitors) because of moderate or severe chronic GvHD.

### 3.4. The Prevalence of VZV Reactivation in the Entire Group Before Vaccination

VZV reactivation was diagnosed in 49 out of 149 (33%) patients, including 5 (3%) patients with reactivation within the first year after transplantation and the remaining 44 (30%) within the subsequent three years.

At this time, the majority of patients no longer received acyclovir prophylaxis, including five patients who stopped recommended prophylaxis despite receiving immunosuppressive treatment (three due to chronic GvHD, two for its prevention). The most common clinical manifestation of VZV reactivation involving intercostal nerves was diagnosed in 40 (81%) patients. The remaining patients had unusual locations, including three patients with cranial nerve involvement, two with ophthalmicus, two with ulnar nerves, one with sacral plexus involvement, and one with a disseminated form of herpes zoster (HZ, diagnosed in a patient with Wiskott–Aldrich syndrome). Four patients required hospitalization: one due to disseminated herpes zoster, one due to ophtalmicus, and two others for infectious complications (1—pneumonia, 1—bronchitis). Postherpetic neuralgia was an essential complication in many of them. There was no intercurrent infection with Epstein–Barr virus or cytomegalovirus in patients with HZ, but in two of them, shingles appeared 24 h after the SARS-CoV2 mRNA vaccine. Figure 1 shows the time from allo-HSCT to VZV reactivation in unvaccinated and vaccinated cohorts. Table 2 summarizes the course of HZ in unvaccinated patients.

### 3.5. The Assessment of the Vaccinated Group

Twenty-one recipients (median age: 41) received two doses of RZV (median time 34 months after transplantation (range, 12–84 months)). Patients were vaccinated when the vaccine became available, which resulted in variability in time duration between transplantation and vaccination. There were 11 seronegative, 2 equivocal, 1 seropositive, and 7-unassessed patients before vaccinations (all with a history of chickenpox in childhood). Eighteen of them had been vaccinated at an interval of 1 month, and the remaining three at an interval of 2 months. Several patients complained about mild pain, erythema, swelling, or fatigue—CTCAE grade 1 after injection.

At the time of vaccination, four patients had been receiving chronic immunosuppressive treatment due to severe (three patients) or moderate (one patient) chronic cGvHD. The serological post-vaccination response (measured 2–3 months after vaccination) was confirmed in 12 (57%) recipients, with a ratio of 2.38–8.3 (median 5.095). Despite vaccination, four patients developed HZ. These patients were not receiving immunosuppressive therapy at the time of disease onset. One of them, who initially had myelodysplastic syndrome, had received rituximab a year earlier for PRCA (pure red cell aplasia) after transplantation. Three others underwent transplantation for lymphoid malignancies (two—B-cell ALL, one—DLBCL), including one who required IgG supplementation until 3 months before disease onset. Three of these patients had a positive serological response to vaccinations.

The median counts of total blood cells in the vaccinated group 1–4 months before vaccination were 451 (range 309–1139) for CD3+4+, 855 (range 137–823) for CD3+8+, 406 (range 70–460) for CD19+, and 211 (range 80–430) for NK cells. None of the vaccinated patients had decreased counts of total CD3+8+, or CD19+ below LLN. All but three (86%) had a lower CD4/CD8 ratio. Six vaccinated patients had absolute counts of total CD3+CD4+cells below LLN, including two who developed clinical disease. Three vaccinated patients had absolute counts of total NK cells below LLN, including one with clinical disease. The vaccinated patients’ characteristics are shown in Table 3.

## 4. Discussion

Secondary immunodeficiencies observed in HSCT recipients increase the risk of VZV reactivation, according to the Hope-Simpson hypothesis that a person’s immune status determines their likelihood of developing HZ [1]. Pre-, peri-, and post-transplant factors, especially multi-stage immune reconstitution, lead to humoral and cellular deficiencies that may promote the occurrence of HZ.

Another critical issue may be immunosenescence, a documented factor that causes virus-specific cellular immunity to wane with age [2,3,4]. Faster-aging lymphoid-biased HSCs (Ly-HSCs) in post-HSCT do not efficiently generate lymphoid progeny and can weaken VZV-specific CMI (T-cell-mediated immune) response [15,16,17,18,19].

The first objective of our study was to assess the incidence of HZ in recipients after HSCT, and VZV reactivation was diagnosed in 33% of patients within the first four years after transplantation. Apparently, acyclovir prophylaxis during the first year post-transplantation, used according to the EBMT guidelines, reduced the frequency of VZV reactivation during this time. Still, the frequency of the disease in the later period after transplantation (without prophylaxis) was a significant challenge [7,12].

According to the literature data, a delayed increase in VZV reactivation may be explained by the underlying immunosuppression rather than a possibility of a “rebound” effect. However, randomized studies suggest that subclinical VZV reactivation (endogenous and exogenous amplification) continues independently of acyclovir prophylaxis and that antigen exposure is sufficient for VZV-specific immune reconstitution [20,21].

The frequency of VZV reactivation varies depending on the transplant center; it was reported to be between 20% and 53%, with an increasing frequency in subsequent years (5% in the first year after HSCT, 21% in the second year, 22.9% in the third year, and 37% in the fifth year). This rate of VZV reactivation in the analyzed group of HSCT recipients is significantly increased compared to people of similar age in the non-HSCT population, where the frequency is reported to be only 7–8‰ [1,4,22].

Nineteen percent of our patients had unusual clinical manifestations of VZV reactivation, including two who developed ulnar nerve involvement after COVID-19 vaccination. VZV reactivation is a known potential adverse event for all COVID-19 vaccines [23]. However, manifestation involving dermatomes of the vaccinated arm has not yet been described.

Four (8.1%) of our patients required hospitalization, a significantly higher number compared to the non-HSCT population (0.05‰) [22]. One of these cases was a man initially suffering from primary immunodeficiency (PID) who developed severe disseminated HZ. It is known that PID recipients may experience poorer T- and B-cell reconstitution after allo-HSCT, which may complicate the course of the disease [19,24].

It is also worth mentioning that having shingles exposes this vulnerable group of patients to the risk of long-term transplant complications, such as increased risk of secondary cancer, stroke, myocardial infarction, and, crucially, persistent postherpetic neuralgia [24,25,26,27,28,29,30,31,32,33,34].

The second part of our study was devoted to the problem of vaccination with the recombinant, adjuvanted zoster vaccine.

The efficacy of this vaccine in the analyzed group, measured by the number of HZ occurrences after vaccination, was over 80%. There are not many data on the effectiveness of the recombinant vaccine against herpes zoster after allo-HSCT. In a randomized clinical trial of auto-HSCT recipients, the efficacy of vaccines was 63.8%. In contrast, in a single-center prospective study of allo-HSCT recipients published by Baumrin et al., it was 97.5% [27]. The comparison of the obtained results is difficult, considering the different vaccination schedules and different elapsed times between transplantation and the start of vaccination.

It is essential to determine the optimal time to administer vaccinations, considering immune reconstitution, especially the counts of CD4+ and CD8+ effector and memory T cells (because of their importance in T-cell-mediated VZV-immune (CMI) responses). None of the analyzed vaccinated patients had decreased counts of total CD3+8, whereas six of the CD3+4+ counts were diminished. Among four patients who developed HZ, two had absolute CD3+CD4+ counts below the LLN, and one had a deficiency of NK cells.

According to the literature, detection of VZV memory T cells is usually possible by 9 to 12 months after HCT, while VZV recurrence correlates significantly with their deficiency [28].

The potential clinical efficacy of VZV subunit vaccines containing glycoprotein E in HCT recipients is related to driving VZV-specific CD4+ and CD8+ T-cell reconstitution [4]. Therefore, having a CD4+ T-cell count ≥200 cells/μL may be necessary for successful vaccination [4].

An insufficient count of NK cells may also have great importance in the susceptibility to HZ. Nonspecific antiviral immunity, especially via IFN-α and granulysin made by NK cells, has direct antiviral activity against VZV and enhances the early destruction of VZV-infected cells [28].

In our analysis, we used serological testing to check post-vaccination responses, aware that no data for immunocompromised patients could guide cut-offs for positive antibody titers. The serological post-vaccination response was confirmed in 12 (57%) recipients, including 4 who developed overt disease despite vaccinations. Three of these four patients initially had lymphoid malignancies before transplantation, and the fourth had been treated with rituximab after allo-HSCT (due to pure red cell aplasia). According to the literature, patients with B-cell malignancies are particularly vulnerable to total and/or functional post-transplant hypogammaglobulinemia, which results from disease-related effects and treatment-related side effects [27,30]. Functional immunoglobulin deficiencies in these recipients may play a role in the expected efficacy of vaccinations in post-transplant care. However, results of a prospective trial by Jotschke et al. suggests that patients with hematological malignancies with insufficient humoral responses might still benefit from vaccinations through cellular responses (regarding the SARS-CoV-2 vaccine) [35]. The importance of cellular response to vaccinations following allo-HSCT requires further investigations.

It is worth mentioning that four vaccinated patients with cGvHD did not develop clinical zoster disease. They were in the chronic phase, on stable, partially reduced doses of immunosuppressive therapy, and all had regenerated lymphocyte subpopulations.

Due to the small group of recipients (RZV was not reimbursed for allo-HSCT recipients in Poland at that time) and the short follow-up period, our results are preliminary. Still, they may contribute to the discussion around the validity and effectiveness of vaccinations against VZV in these SID patients.

## 5. Conclusions

Herpes zoster is a common complication after allo-HSCT and occurs in over 30% of recipients, mainly in the late period after transplantation after completion of acyclovir prophylaxis. The preliminary results indicate that RZV vaccination after allo-HSCT was safe and more than 80% effective at preventing HZ, but some vaccinated individuals did experience HZ.

## Figures and Tables

**Figure 1 idr-17-00048-f001:**
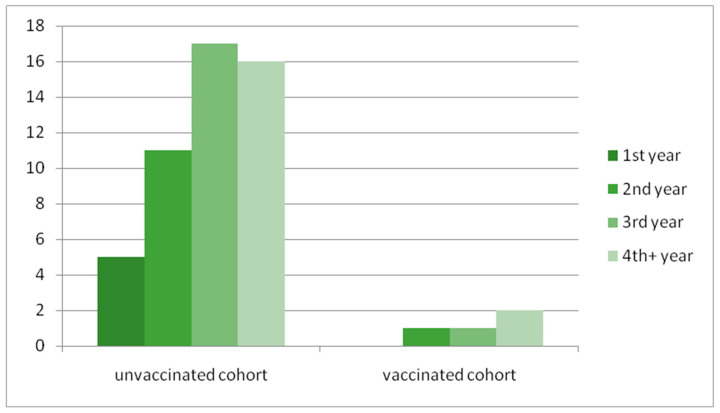
The time from allo-HSCT to VZV reactivation in unvaccinated and vaccinated cohorts.

**Table 1 idr-17-00048-t001:** Patients’ baseline characteristics.

	Number of Patients	Share of the Total (%)	Number of Patients Who Experienced HZ	Share of All HZ Patients (%)	Share of HZ Patients of the Total (%)
Gender					
Male	85	57	30	61	20
Female	64	43	19	39	12
Age in years					
18–40	57	38	24	49	16
41–60	74	49	20	41	13
>=60	18	13	5	10	3
Diagnosis					
AML	82	55	27	55	18
MDS	17	11.5	7	14	5
ALL	17	11.5	6	12	4
AA	6	4	3	6	2
MPN	14	9	5	10	3
Lymphoma	13	9	1	2	0.7
Conditioning					
MAC	104	70	37	76	25
NMA	5	3	3	6	2
RIC	40	27	9	18	6
Donor					
MRD	49	33	15	31	10
MUD	83	56	30	61	20
MMUD	12	8	3	6	2
Haploidentical	5	3	1	2	0.7
Acute GvHD					
Grade 1–2	40	27	2	4	1
Grade 3–4	24	16	6	12	4
Chronic GvHD					
Mild	17	11	15	30	20
Moderate	33	22	6	12	4
Severe	34	34	4	8	4

(AA—aplastic anemia, AML—acute myeloid leukemia, ALL—acute lymphoblastic leukemia, MAC—myeloablative conditioning, MDS—myelodysplastic syndrome, MPN—myeloproliferative neoplasm, MMUD—mismatched unrelated donor, MRD—matched related donor, MUD—matched unrelated donor, NMA—non-myeloablative conditioning, RIC—reduced-intensity conditioning).

**Table 2 idr-17-00048-t002:** The summary of the course of HZ in unvaccinated patients.

Clinical Manifestation	Number/Share of the Total	Need for Hospitalization
Intercostal nerves	40 (81%)	
Cranial nerve	3 (6%)	
Ophtalmicus	2 (4%)	1
Ulnar	2 (4%)	
Sacral plexus	1 (2%)	
Disseminated form	1 (2%)	1
Bacterial complication	2 (4%)	2

**Table 3 idr-17-00048-t003:** Vaccinated patients’ characteristics (patients who developed herpes zoster are marked in bold; patients treated with immunosuppressive therapy due to GvHD at the time of vaccination are marked in italics).

	Age	Gender	Diagnosis	Time to First Vaccine Dose After HSCT (Months)	Ratio Value for VZV IgG Before and After Vaccination (or *Titer in mIU/mL)		Time From the First Vaccination to HZ, Clinical Course	CD4/µL N [309–1139]	CD8/µL N [137–823]	CD19/µL N [70–460]	NK/µL N [80–430]	CD4/CD8 N [1.0–5.0]
					Before	After						
1	22	F	AML	12	0.64	7.07		616	535	227	211	1.2
2	34	M	AML	18	0.52	8.16		220	642	406	406	0.3
3	53	F	CML	36	1.15	7.43		135	450	83	45	0.3
**4**	**40**	**F**	**MDS**	**31**	No data	**2.4**	6 months, intercostal HZ with bacterial complications	**266**	**931**	**418**	**228**	**0.3**
5	33	F	MDS	35	0.25	4.28		672	496	160	208	1.4
**6**	**41**	**M**	**DLBCL**	**18**	**0.77**	**3.09**	5 months, disseminated form of HZ	**360**	**1400**	**1240**	**840**	**0.3**
*7*	*38*	*M*	*PTCL*	*60*	*No data*	*4.05*		*1161*	*826*	*258*	*258*	*1.4*
8	50	M	CML	14	142 *	6.86		475	1425	1536	144	0.3
*9*	*54*	*F*	*AML*	*84*	*No data*	*3.44*		*312*	*396*	*408*	*72*	*0.8*
10	67	F	AML	61	No data	3.16		260	900	No data	No data	0.3
11	50	M	AML	34	0.65	3.92		315	1095	No data	No data	0.3
12	67	M	AA	48	No data	5.91		223	594	127	106	0.4
13	22	F	AML	35	0.98	2.38		419	855	346	164	0.5
14	36	F	ALL	29	0.72	No data		469	2180	110	83	0.2
15	52	F	AML	12	0.44	No data		658	2632	799	423	0.3
16	22	F	AML	17	0.7	8.3		451	354	No data	No data	1.27
*17*	*48*	*F*	*AML*	*48*	*No data*	*No data*		*718*	*800*	*773*	*359*	*0.9*
*18*	*66*	*M*	*MDS*	*14*	*No data*	*1041 **		*654*	*2755*	*607*	*467*	*0.2*
**19**	**23**	**M**	**ALL**	**47**	**1.01**	**7.57**	10 months HZ with cranial nerve involvement	**266**	**714**	**336**	**42**	**0.4**
**20**	**26**	**F**	**ALL**	**26**	**90 ***	**538 ***	9 months, intercostal form of HZ	**783**	**1276**	No data	No data	**0.6**
21	66	F	AML	39	0.86	6.39		842	1427	421	356	0.6

(AML—acute myeloid leukemia, CML—chronic myeloid leukemia, MDS—myelodysplastic syndrome, DLBCL—diffuse large B-cell lymphoma, PTCL—peripheral T-cell lymphoma, ALL—acute lymphoblastic leukemia, AA—aplastic anemia).

## Data Availability

The data presented in this study are available on request from the corresponding author due to privacy.

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
