# Peer review of "Prevalence of VZV Reactivation and Effectiveness of Vaccination with Recombinant Adjuvanted Zoster Vaccine in Allogeneic Hematopoietic Stem Cell Recipients—A Single-Center Analysis"

_2036-7449, 2025, doi:10.3390/idr17030048_

Round 1

Reviewer 1 Report

Comments and Suggestions for Authors

In this manuscript, Ewa Karakulska-Prystupiuk and colleagues describe the prevalence of VZV reactivation in allogeneic hematopoietic stem cell recipients as well as the efficacy of a VZV vaccination in a small subgroup of these patients. The manuscript is informative but there are a few issues that need to be clarified:

  1. There are some data missing in Table 2. The CD4/CD8 ratio should be easy to calculate for patient 16 given that you have CD4 and CD8 values. If data are missing for NK cells and B-cells, please indicate this in the table (“data missing”).
  2. Table 2: Please explain how “antibody ratio” and “U/ml” relate to each other.
  3. It is not clear what the follow-up time was for VZV reactivation in the unvaccinated patients.
  4. VZV reactivation occurred within the first 4 years after transplantation in the unvaccinated cohort. How long time after transplantation did VZV reactivate in the vaccinated cohort? Please construct a graph of “time to reactivation (time to event)” for the unvaccinated and the vaccinated patients to clarify this.
  5. Please be consistent with your headings in the results section. Some are numbered and in italics, others are not numbered and in bold.

Author Response

Reviewer 1 Response:

We sincerely appreciate your valuable time and effort in reviewing our manuscript.

We have addressed each point and provided a point-by-point response below, outlining the revisions made.

Reviewer 1 comment 1;

There are some data missing in Table 2. The CD4/CD8 ratio should be easy to calculate for patient 16 given that you have CD4 and CD8 values. If data are missing for NK cells and B-cells, please indicate this in the table (“data missing”).

[Response] Thank you for your suggestion.

We calculated the missing CD4/8 count for patient 16 and indicated "data missing" in the table.

Reviewer 1 comment 2;

Table 2: Please explain how “antibody ratio” and “U/ml” relate to each other.

[Response] Thank you for bringing up this important consideration

These two methods measure antibody levels differently and are not directly interchangeable.

The comparison of results of these two tests may only be qualitative because the ELISA ratio is a relative value and is not standardized to the WHO international unit.

We have added this clarification to the Methods section 2.4, as follow:

The anti-VZV (IgG) ELISA test manufactured by Euroimmun was primarily used, with the result expressed as a ratio (the optical density (OD) of the patient sample divided by the OD of cut-off).

 A ratio≤0.8 was considered negative, 0.8-1.1 equivocal, and > 1.1 positive.

Direct chemiluminescent immunoassay (CLIA) by DiaSorin (analyzer LIAISON XL) was used in three patients, with quantitative results expressed in International Units per milliliter (IU/ml). A titer≥ 150mIU/ml was considered a positive result.

The comparison of the results of these two tests may only be qualitative because the ELISA ratio is a relative value and is not standardized to the WHO international unit.

Reviewer 1 comment 3;

It is not clear what the follow-up time was for VZV reactivation in the unvaccinated patients.

[Response] Thank you for your insightful observation.

The follow- up time for VZV reactivation in unvaccinated patients was the first eight years after transplantation.

We have added this clarification to the Methods; Study population section 2.1

Reviewer 1 comment 4;

VZV reactivation occurred within the first 4 years after transplantation in the unvaccinated cohort. How long time after transplantation did VZV reactivate in the vaccinated cohort? Please construct a graph of “time to reactivation (time to event)” for the unvaccinated and the vaccinated patients to clarify this.

[Response] Thank you for your suggestion.

We have added the requested graph below. We decided to name the last column as the "4th+  year", because in later years the outpatient follow-up was less frequent.

Figure 1. The time from allo HSCT to VZV reactivation

Reviewer 1 comment 5;

Please be consistent with your headings in the results section. Some are numbered and in italics, others are not numbered and in bold.

 [Response] Thank you for your suggestion. We have improved the layout.

We appreciate the reviewer’s detailed feedback, which has greatly improved our manuscript. We have carefully addressed all comments and made the necessary revisions accordingly. We hope that the revised manuscript meets your expectations and look forward to your further evaluation.

Reviewer 2 Report

Comments and Suggestions for Authors

This work from Karakulska-Prystupiuk and colleagues describes a cohort of patients that have undergone allogenic haematopoietic stem cell transplantation (HSCT), and is particularly focused on the frequency of varicella zoster (VZV) reactivation following transplantation. They also report some data on patients that have received the reasonably recently developed VZV glycoprotein E vaccine. The clinical data described is useful and will be of value to researchers and clinicians.

One general comment is a lack of references in the introduction and discussion. Whilst many of the statements may be broadly understood to be true, some reference to pertinetent literature should be included to back it up (e.g. lines 60, 61, 64-65, 249-252).

In table 1, the authors summarise the characteristics of their unvaccinated patient cohort. It would be valuable for the authors to indicate how many members of each population ultimately experienced a VZV reactivation event. This would add to the body of data available on this and might help target future interventions towards  specific patient subgroups. The authors could also include an additional small table summarising relevant virological parameters related to the patients who experienced HZ (average/range of months post transplantation that onset occurred, duration, hospitalisations etc), which would expand the dataset and make it easier to retrieve the information that is currently contained only within the text.

In table 2, the authors provide data on each individual patient that received VZV vaccination post-transplantation. The authors could add the time post vaccination until HZ onset for the 4 patients who experienced it. Information on the severity of HZ experienced by vaccinated individuals would also be valuable, as whilst the vaccine may not have prevented HZ in these cases, reduction of severity of symptoms is still a valuable function.

One thing that should be noted is there is no attempt to provide any  formalised statistical analysis of the data. The vaccinated patient group is certainly too small to provide robust statistical data, and consequently whilst the statement used by the authors in the abstract and summary (‘may have limited effectiveness) is very gentle, I would recommend its removal. In fact, according to this same data, the vaccine is more than 80% effective at preventing HZ in this very immunologically challenged population, which could be considered highly effective! Perhaps a more directly factual ‘vaccination was >80% effective at preventing HZ but some vaccinated individuals did experience HZ’, or words to that effect.

The authors also note that in their limited number of patients, patients transplanted due to lymphoid malignancies were perhaps less able to respond to VZV vaccination. There seems to be some evidence that this is a more generalisable trend to other vaccinations, but particularly to humoral responses, whilst cellular responses were well generated. The discussion element for this could be broadened, and be used as an opportunity to call for investigation of the development of cellular responses to vaccination in allo-HSCT patients.

Minor edits:

Line 93: Letermovir dosage should presumably be 240-480mg

Line 150: Misformatting of the word ‘one’

Table 1: Replace ‘Haplo’ with ‘Haploidentical’

Line 108-109: Was this a commercial CLIA assay that can be referenced more clearly?

Comments on the Quality of English Language

The quality of the English language is generally quite good. It is understandable, but there are some minor errors related to tense or precise sentence construction, along with some minor formatting errors additional to those mentioned above. This makes a few areas a little harder to process and not flow so smoothly. 

Author Response

Reviewer 2

We sincerely appreciate the reviewer’s insightful comments and constructive feedback. We have addressed each point and provided a  point-by-point response below, outlining the revisions made.

Reviewer 2 comment 1;

One general comment is a lack of references in the introduction and discussion. Whilst many of the statements may be broadly understood to be true, some reference to pertinetent literature should be included to back it up (e.g. lines 60, 61, 64-65, 249-252).

[Response] Thank you for your suggestion. We have included additional references.

Reviewer 2 comment 2;

In table 1, the authors summarise the characteristics of their unvaccinated patient cohort. It would be valuable for the authors to indicate how many members of each population ultimately experienced a VZV reactivation event. This would add to the body of data available on this and might help target future interventions towards  specific patient subgroups. The authors could also include an additional small table summarising relevant virological parameters related to the patients who experienced HZ (average/range of months post transplantation that onset occurred, duration, hospitalisations etc), which would expand the dataset and make it easier to retrieve the information that is currently contained only within the text.

[RESPONSE] We have completed Table 1 according to the instructions and added a new table 3. summarizing the course of HZ in unvaccinated patients, except for the time of disease onset, which was presented graphically (according to the recommendations of the first reviewer).

Reviewer 2 comment 3;

In table 2, the authors provide data on each individual patient that received VZV vaccination post-transplantation. The authors could add the time post vaccination until HZ onset for the 4 patients who experienced it. Information on the severity of HZ experienced by vaccinated individuals would also be valuable, as whilst the vaccine may not have prevented HZ in these cases, reduction of severity of symptoms is still a valuable function.

[RESPONSE] We have completed Table 1 according to the instructions.

Reviewer 2 comment 4;

One thing that should be noted is there is no attempt to provide any  formalised statistical analysis of the data. The vaccinated patient group is certainly too small to provide robust statistical data, and consequently whilst the statement used by the authors in the abstract and summary (‘may have limited effectiveness) is very gentle, I would recommend its removal. In fact, according to this same data, the vaccine is more than 80% effective at preventing HZ in this very immunologically challenged population, which could be considered highly effective! Perhaps a more directly factual ‘vaccination was >80% effective at preventing HZ but some vaccinated individuals did experience HZ’, or words to that effect.

[RESPONSE] We have changed the abstract’s conclusion and the summary as suggested.

Reviewer 2 comment 5;

The authors also note that in their limited number of patients, patients transplanted due to lymphoid malignancies were perhaps less able to respond to VZV vaccination. There seems to be some evidence that this is a more generalisable trend to other vaccinations, but particularly to humoral responses, whilst cellular responses were well generated. The discussion element for this could be broadened, and be used as an opportunity to call for investigation of the development of cellular responses to vaccination in allo-HSCT patients.

[RESPONSE] We have included one reference – item 35 and changed this part of the discussion as follows; Functional immunoglobulin deficiencies in these recipients may play a role in the expected efficacy of vaccinations in post-transplant care. However, results of a prospective trial by Jotschke et al. suggests that patients with hematological malignancies with insufficient humoral responses might still benefit from vaccinations through cellular responses (regarding SARS-CoV-2 vaccine)[35]. The importance of cellular response to vaccinations following allo-HSCT requires further investigations.

Reviewer 2 comment 6;

Minor edits:

Line 93: Letermovir dosage should presumably be 240-480mg

Line 150: Misformatting of the word ‘one’

Table 1: Replace ‘Haplo’ with ‘Haploidentical’

Line 108-109: Was this a commercial CLIA assay that can be referenced more clearly?

[RESPONSE] We have corrected the indicated errors and detailed the CLIA test commercial data.

We appreciate the reviewer’s detailed feedback, which has greatly improved our manuscript. We have carefully addressed all comments and made the necessary revisions accordingly. We hope that the revised manuscript meets your expectations and look forward to your further evaluation.
